# An automated cirrus classification

Edward Gryspeerdt[1,2], Johannes Quaas[2], Tom Goren[2], Daniel Klocke[3], and Matthias Brueck[4]

[1]Space and Atmospheric Physics Group, Imperial College London, London, United Kingdom
[2]Institute for Meteorology, Universität Leipzig, Leipzig, Germany
[3]Hans-Ertel-Zentrum für Wetterforschung, Deutscher Wetterdienst, Offenbach, Germany
[4]Max-Planck-Institut für Meteorologie, Hamburg, Germany

*Correspondence to:* E. Gryspeerdt
(e.gryspeerdt@imperial.ac.uk)

**Abstract.** Cirrus clouds play an important role in determining the radiation budget of the earth, but many of their properties remain uncertain, particularly their response to aerosol variations and to warming. Part of the reason for this uncertainty is the dependence of cirrus cloud properties on the cloud formation mechanism, which itself is strongly dependent on the local meteorological conditions.

In this work, a classification system (Identification and Classification of Cirrus or IC-CIR) is introduced to identify cirrus clouds by the cloud formation mechanism. Using re-analysis and satellite data, cirrus clouds are separated in four main types: orographic, frontal, convective and synoptic. Through a comparison to convection-permitting model simulations and back-trajectory based analysis, it is shown that these observation-based regimes can provide extra information on the cloud scale updraughts and the frequency of occurrence of liquid-origin ice, with the convective regime having higher updraughts and a greater occurrence of liquid-origin ice compared to the synoptic regimes. Despite having different cloud formation mechanisms, the radiative properties of the regimes are not distinct, indicating that retrieved cloud properties alone are insufficient to completely describe them.

This classification is designed to be easily implemented in GCMs, helping improve future model-observation comparisons and leading to improved parametrisations of cirrus cloud processes.

## 1 Introduction

High clouds are a key component of the Earth's energy budget, although there is still considerable uncertainty about cloud formation mechanisms and their response to environmental changes, particularly in response to changes in the aerosol environment (Heyn et al., 2017) and to warming (Bony et al., 2016). Thin cirrus clouds tend to have a positive cloud radiative effect (warming the atmosphere)(Chen et al., 2000). Their radiative properties are strongly controlled by their altitude and water content in addition to their microphysical properties, particularly the ice water path (IWC), ice crystal number concentration ($N_i$) and the crystal size distribution (Fu and Liou, 1993). Constraining the impact of cloud controlling factors on cirrus clouds is thus vital to improve the parametrisation of cirrus clouds and to constrain their response to aerosol and warming.

Similar to the effect of aerosols on liquid clouds, an aerosol influence on ice clouds would likely modify ice nucleation processes, changing the $N_i$, perhaps by orders of magnitude (Kärcher and Lohmann, 2003; Heymsfield et al., 2017) and impacting the cloud development. Ice nucleation mechanisms and rates are strongly temperature dependent. At temperatures warmer than about -37.5°C, ice crystals are formed either by heterogeneous nucleation from ice nucleating particles (INP) or through the freezing of liquid droplets (via either immersion or contact freezing) with the INP concentration providing a strong constraint on $N_i$ formed from heterogeneous nucleation. At temperatures colder than around -37.5°C, ice can also nucleate homogeneously (without an INP), through either the freezing of liquid aerosol or the freezing of remaining liquid droplets. These processes are dependent on the supersaturation, with homogeneous nucleation being restricted to higher supersaturations than heterogeneous nucleation. The relative importance of these different processes is relevant for determining the $N_i$ and the ice crystal size distribution (e.g. Heymsfield et al., 2017; Kärcher, 2017), which can affect the reflectivity, extent and lifetime of a cirrus cloud.

The ice nucleation rate and the nucleating ability of INP is thus a strong function of temperature and of supersaturation (Hoose and Möhler, 2012), which is in turn related to the strength of the cloud-scale updraughts. These factors vary by cloud type. With very high updraught speeds, convective clouds can contain liquid water to temperatures as low as -37°C (Rosenfeld, 2000), suggesting an important role of liquid origin ice. In contrast, tropical tropopause cirrus are more likely to contain ice formed in-situ (e.g. Jensen et al., 2010). The importance of the origin of the ice in a cloud (liquid or in-situ) has recently been introduced and demonstrated by Krämer et al. (2016); Luebke et al. (2016), using in-situ observations to show that liquid origin cirrus typically have higher IWC and $N_i$ than in-situ formed cirrus.

Understanding how cirrus clouds and the $N_i$ in particular respond to these four factors (temperature, in-cloud updraught, liquid/in-situ origin and aerosol environment) is vital for improving cloud parametrisations in atmospheric models. Temperature and the aerosol environment can be determined from reanalysis data (Dee et al., 2011; Benedetti et al., 2009; Morcrette et al., 2009). However, the cloud scale updraught and the ice origin are not often directly simulated in atmospheric models, being calculated through parametrisations. As such, reanalysis values of these quantities may not be suitable for use as a constraint on cloud ice microphysics parametrisations in general circulation models (GCMs). Developing a classification for cirrus clouds that can provide information on the in-cloud updraught and the ice origin is the main focus of this work.

## 1.1 Existing classifications

The most common classification of cirrus clouds is based on their surface observed properties, based on the work of Howard (1803) and formalised by the World Meteorological Organization (2017). Although this classification can be easily applied by surface observers, a lack of data availability in many regions of the globe (Woodruff et al., 2011) and the obscuring of cirrus from the surface by low cloud means that there are significant advantages to a satellite-based classification. The manual classifications that have been used in past studies (e.g. Sassen and Comstock, 2001) are labour intensive, making them difficult to apply to large satellite datasets. An automated classification based on satellite data and reanalysis data is thus a necessary step forward in order to provide observational constraints on cirrus cloud processes for large statistical analyses.

Existing automated cloud classifications can be grouped into two broad categories, although both categories are often mapped to the Howard (1803) classification. "Cloud regimes" are based on the observed properties of clouds. As an example, Inoue (1987) determined a regimes classification based on satellite brightness temperatures allowing a separation of convective cloud cores from the anvils that surround them. More recent studies have used extra cloud properties in defining a regime classification (such as cloud optical depth), producing cirrus cloud regimes in addition to regimes describing low cloud properties (e.g. Rossow et al., 2005; Gryspeerdt and Stier, 2012; Tselioudis et al., 2013; Oreopoulos et al., 2016). These methods require that the properties defining the cloud regimes vary strongly between the cloud types, but this also means that if the cloud properties used to define the regimes change (perhaps as a function of aerosol), this will change both the properties and the occurrence of the regimes (Williams and Webb, 2009; Gryspeerdt et al., 2014). Being based purely on the observed cloud properties, "cloud regimes" do not require assumptions about the impact of local meteorology on the regime occurrence.

Conversely, "dynamical regimes" are based on the meteorological situation, often using reanalysis data as a method of defining the regimes (e.g. Medeiros and Stevens, 2009; Muhlbauer et al., 2014). These regimes can often be related to specific cloud types, but they are not necessarily a good constraint on the cloud properties if the regimes are not defined using the correct parameters (Nam and Quaas, 2013; Leinonen et al., 2016). However, it is not required that these regimes map to the "Howard" classification. For example, Wernli et al. (2016) use reanalysis data to classify cirrus as either liquid or in-situ origin, depending on their meteorological history and the parametrised cloud phase within the reanalysis. "Dynamical regimes" require assumptions to be made about the important meteorological variables and may have to rely on the accuracy of reanalysis products. However, as a change in the properties of a cloud does not change the occurrence of a "dynamical regime", using "dynamical regimes" can simplify analyses into the response of cloud parameters to meteorological parameters not used for defining the regimes.

In this work, elements of both the "cloud regime" and "dynamical regime" methods are combined to develop a source-based classification of cirrus and other high cloud, using satellite and reanalysis data. The aim of this classification is to provide information on the cloud-scale updraughts and ice origin within cirrus clouds. The classification will be compared against the Wernli et al. (2016) classification and convection-permitting simulations from the ICON (ICOsahedral Non-hydrostatic) model (Zängl et al., 2015) to examine how much information it provides on the ice origin and the cloud-scale updraughts. By separating frequency of occurrence of cloud-forming meteorological conditions from the properties of the clouds that form as a result (similar to "dynamical regimes"), this classification also aims to improve process-based observational comparisons with GCMs. This will enable future studies to combine the regimes defined here with reanalysis temperature and aerosol properties, along with additional observational data to investigate the controls on ice nucleation processes and cirrus cloud properties.

## 2 Methods

For this classification, the regimes are derived from four main sources of cirrus cloud: orographic uplift, frontal uplift, convective systems and synoptic cirrus formed through large scale rising motions. These are divided into the twelve cirrus cloud regimes specified in Tab. 1.

| ID | Short name | Class | Definition |
|---|---|---|---|
| 11 | O2 | Oro. 2 | Highest sextile of orographic updraught |
| 10 | O1 | Oro. 1 | Second sextile of orographic updraught |
| 9 | F | Frontal | Within "blob" of high cloud that intersects a reanalysis front |
| 8 | F2 | Frontal 2deg | Within $2°$ of frontal/F |
| 7 | F5 | Frontal 5deg | Within $5°$ of frontal/F |
| 6 | C | Convective | Within a blob intersecting a region of negative $\omega(500\,\text{hPa})$ |
| 5 | C2 | Conv. 2deg | Within $2°$ of conv./C |
| 4 | C5 | Conv. 5deg | Within $5°$ of conv./C |
| 3 | J | Jet | Windspeed$(300\,\text{hPa}) > 30\,\text{ms}^{-1}$ |
| 2 | Su | Synoptic up | Negative $\omega(500\,\text{hPa})$ |
| 1 | Sd | Synoptic down | $\omega(500\,\text{hPa}) < 0.05\,\text{Pa}\,\text{s}^{-1}$ |
| 0 | S | Synoptic | None of the above |

**Table 1.** Classification criteria. Gridboxes are assigned the highest applicable class, regardless of if a cloud is detected.

.

Each $1° \times 1°$ gridbox globally is assigned to one of these regimes, irrespective of whether a cloud is observed. This ensures that every location is assigned to a regime, such that the regime occurrence is not biased by any satellite cloud detection threshold. This also enables the occurrence of the meteorological conditions to be separated from the properties of the clouds that occur within each regime, making the classification more suitable for a process-based analysis of the cirrus clouds in GCMs.

As an aim of this work is to generate a classification that is applicable to GCMs, consideration is given to the data volume that would be required to generate the classification and availability of diagnostics and observational measurements. As such, only a two dimensional classification is created. Gridboxes are classified in the following order, with the first set of criteria that are satisfied (Tab. 1) determining the regime.

1. Orographic clouds (O1, O2)

   Similar to parametrisations for in-cloud updraughts in orographic clouds (Lott, 1999; Joos et al., 2008), the in-cloud updraught for an orographic cloud is assumed to be proportional to the product of the windspeed at $850\,\text{hPa}$ (or the surface if it is a higher altitude) and the surface topography variation (the difference between the mean and minimum altitude within each $1°\times1°$ degree gridbox). The windspeed is from ERA-Interim (Dee et al., 2011) and the topographic data is from the United States Geological Survey GMTED2010 dataset gridded to $0.1°\times0.1°$ resolution.

Gridboxes falling into approximately the upper tercile of parametrised updraughts are assigned to orographic regimes. Those with a windspeed-surface topography variation product of greater than 880 m$^2$s$^{-1}$ are assigned to the O1 regime (Tab. 1), with those with a windspeed-surface topography variation product larger than 1800 m$^2$s$^{-1}$ forming the O2 regime. These constants were selected based on a year of data to give approximately equal relative frequencies of occurrence (RFO) for the O1 and O2 regimes. The orographic regime is defined first, given their dominant control over the in-cloud updraught in mountainous regions (Joos et al., 2008).

2. Frontal clouds (F, F2, F5)

Points are assigned to the frontal regime based on their proximity to atmospheric fronts, located using reanalysis data. Fronts are determined using an objective front detection method based on (Hewson, 1998), using the wet bulb potential temperature ($\theta_w$) at 850 hPa calculated from ERA-Interim reanalysis (Dee et al., 2011) at a 1°×1° resolution following the method from Davies-Jones (2009). The fronts are located using the criterion

$$\nabla \cdot |\nabla |\nabla \theta_w|| = 0 \tag{1}$$

which was shown to provide similar results to a more sophisticated locator based on mean-axes (Hewson, 1998). A single field of fronts is created for a local solar time of 13:30, to align with the A-Train overpass. Fronts which exist for only a single six hour reanalysis timestep are removed, as are fronts that pass through less than ten 1°×1° gridboxes. The quasi-stationary fronts (front speed parameter of less than 1.5 ms$^{-1}$) identified by Berry et al. (2011) are also excluded, as these often occur in regions of tropical convection, where the convective regime is more appropriate.

To identify clouds that are part of a frontal system, cloud "blobs" are created using cloud data from the moderate resolution imaging spectroradiometer (MODIS) instrument on the Aqua satellite (Platnick et al., 2017) at 1° by 1° (MYD08_D3). The "blobs" are defined as contiguous connected regions where the cloud top pressure (CTP) is less than 550 hPa and the optical depth is greater than 5. "Blobs" are not allowed to exist where the orographic regimes have been assigned, nor in regions with topography over 1500 m (approximately 850 hPa), as fronts cannot be accurately determined in these regions. The primary effect of this restriction is to prevent frontal and convective clouds being classified over Greenland and East Antarctica.

These choices ensure that clouds from the same system are placed in the same "blob" while at the same time preventing the formation of a single, global "blob". The requirement for a cloud optical depth retrieval limits these regime definitions to daylight, around 13:30 local solar time (the overpass time of the Aqua satellite on which MODIS is flown).

Clouds are then assigned to the frontal class if they are part of a blob that intersects a front (F). As this method is likely to miss thinner frontal clouds, regions of two (F2) and five (F5) degrees around the edge of the frontal clouds are created to include these clouds. The width of these "buffer" regimes is further considered in the results section.

3. Convective clouds (C, C2, C5)

   Cirrus from convective (non-frontal) clouds is determined using the same "blobs" that are used for the frontal cloud classification. In this case, clouds in a blob that is not labelled as frontal are considered convective (C) if they intersect a region of large scale updraught (as defined by the ECMWF ERA-Interim grid-scale pressure vertical velocity at 500 hPa - $\omega_{500}$). As with the frontal clouds, buffer "regimes" of two (C2) and five (C5) degrees are defined around each convective "blob" to include thinner clouds that are not included in the "blob". The convective regime cirrus are defined after the frontal regimes as many frontal locations also satisfy the criteria for the convective regime.

4. Other classes

   From the remaining pixels, locations with a windspeed at 300 hPa greater than $30\,\mathrm{ms}^{-1}$ are classed as jet-stream cirrus (J). The remaining locations are considered as candidates for synoptic cirrus formation. These are separated into three further regimes using the $\omega_{500}$ to further limit the possibility of convective clouds contaminating the synoptic cirrus regime. Locations with a negative $\omega_{500}$ form the synoptic updraught regime (Su), a positive $\omega_{500}$ less than $0.05\,\mathrm{Pa\,s}^{-1}$ the synoptic weak-updraught (Sd) and the remaining form the final synoptic regime (S). The synoptic regimes are the residual regimes, assigned after the more clearly-defined classes.

Whilst these classes do not cover every cirrus formation mechanism, they are designed to cover the most globally significant sources of cirrus. Other classes or subdivisions could be added in future work.

To examine the ability of this classification for determining cirrus cloud origin, it is compared to the classification from Wernli et al. (2016) over the north Atlantic (30N-80N, 110W-40E) for the year of 2007. The Wernli et al. (2016) classification uses back trajectories in the ECMWF ERA-Interim re-analysis to determine if a cirrus cloud is formed directly in the ice phase, or if it formed through the freezing of liquid droplets. This back-trajectory technique is suitable for use in the regions of large-scale rising air associated with fronts, but the reliance on parametrised convection makes it less suitable to determining the origin in highly convective locations. For each regime, the probability of a trajectory being liquid or in-situ origin at each temperature is calculated, with the same regime assigned at all temperatures within each lat-lon gridbox. As it is based on the ERA-Interim phase parametrisation, it is likely to overestimate the fraction of in-situ origin cases, particularly at temperatures colder than -23°C. This misclassification would not be limited to temperatures warmer than -40°C, as these colder clouds may still be liquid origin cirrus. However, as liquid water is rare at temperatures below -23°C (Choi et al., 2010), the errors phase misclassification are likely small, such that the Wernli et al. (2016) classification is able to provide a useful first comparison for the regimes derived in this work.

Convective clouds are likely produce liquid-origin cirrus (Rosenfeld, 2000), such that the in-cloud updraught is a key parameter for these clouds. The convective regime should be able to identify regions with a higher in-cloud updraught if it is to correctly identify convective origin cirrus. As in-cloud updraughts are not currently retrieved by satellite, the cirrus regimes classification is also compared to output from a convection-permitting simulation using the ICON model (Zängl et al., 2015), performed at 2.5 km resolution over the tropical Atlantic (10S-20N, 68W-15E), with a nested domain at 1.25 km resolution over Barbados (4S-18N, 64W-42W) during August 2016 (see. Fig. 1b). The simulation was initialised each day at 0000 UTC

using the ECMWF operational analysis of the atmospheric state and boundary conditions are provided three hourly from the ECMWF operational forecast. Output from the nearest hour to 1330 LST (the time of the Aqua satellite overpass) across the domain is used to compare with the regimes, corresponding to a period 12-16 hours after the start of the simulation on each day. As the simulation can resolve convection, the updraught velocities in the simulation show whether the cirrus classification 5 is able to provide information on the convective updraught velocities.

To characterise the regimes and guide further studies, the cloud radiative effect (CRE) for each of the regimes are determined following Oreopoulos et al. (2016). Using the CERES SYN1deg daily product at $1°×1°$ resolution (Doelling et al., 2013), daily mean the solar (SW) and terrestrial (LW) CRE is calculated for each of the regimes for the ten year period 2003-2013 inclusive. As the classification is based on assigning high clouds but makes no requirement on any underlying low cloud, the 10 CTP histogram in the MODIS level 3 MYD08_D3 product is used to select gridboxes that have more than 99% of retrieved CTPs higher than 550 hPa, allowing the CRE of the high cloud to be studied separately.

## 3 Results

### 3.1 Example classification

Fig. 1a shows a MODIS true-colour composite from the 12th of April, 2007, Fig. 1b shows the retrieved cloud top pressure 15 for the same day, with the bands of high cloud in the mid latitude and convective systems in the tropics both clearly visible. An examination of the classification (Fig. 1c) shows many of the features commonly seen with this classification method. The frontal cloud fields (F, F2, F5) are clearly visible in bands through the extratropics, although the front detection method does occasionally label frontal clouds in the tropics. The convective cloud regimes (C, C2, C5) occur primarily in the tropics, but can be occasionally seen in relation to frontal clouds. For example, the convection in the cold air outbreak behind the frontal in 20 the southern ocean is clearly visible in Fig. 1. Orographic regimes (O1, O2) are found over land and although they are related to altitude, they are clearly distinct, with the east Antarctica plateau being classed as primarily synoptic cirrus, despite its altitude. This agrees with previous studies showing a low CF over east Antarctica (Bromwich et al., 2012). The jet stream regime (J) is visible in the extratropics, often linking frontal systems.

### 3.2 Relative frequency of occurrence

25 The relative frequency of occurrence (RFO) of the regimes for a period of ten years (Fig. 2) behaves qualitatively similarly to the example day shown in Fig. 1. The RFO of the frontal regimes is highest in the stormtrack regions, with the misclassification in the tropics contributing a small amount to the total RFO of the frontal regime. Similarly, the two and five degree buffer regimes (F2, F5) also show the highest RFO in the extratropical stormtrack regions.

The convective regimes occur primarily in the tropics, although their extra-tropical RFO is not zero. The two and five degree 30 buffer regimes (C2, C5) are more common than the convective regime itself and the five degree buffer regime starts to show a split around the equator.

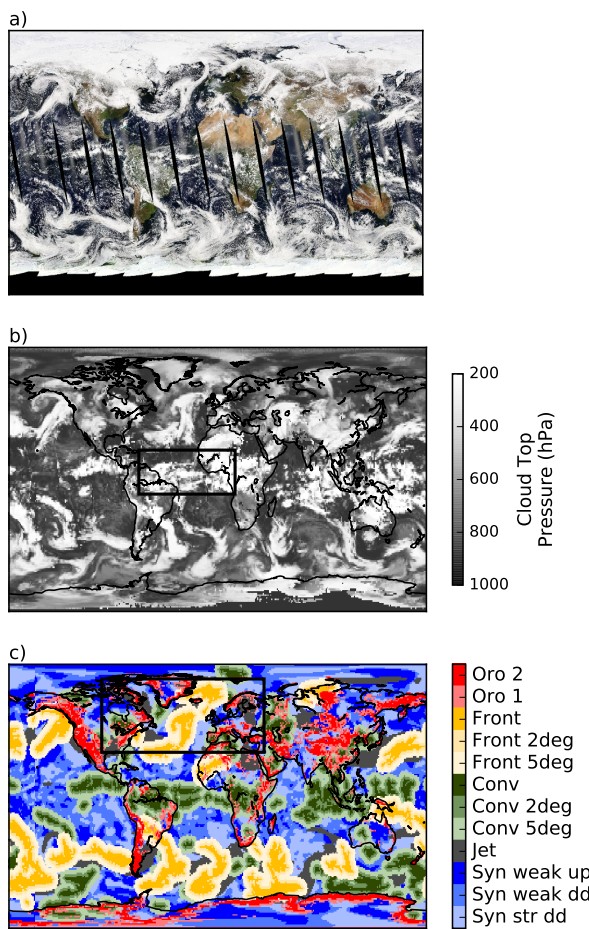

**Figure 1.** a) A MODIS Aqua true-colour image for the 12th of April, 2007. b) The MODIS daytime cloud top pressure, in-filled with the night-time CTP where no daytime retrieval exists. The box in the tropics shows the location of the regional ICON simulation. c) The classification presented in this work, derived from MODIS Aqua, ERA Interim and topographic data for the 12th of April, 2007. The colours denote the classified regime at each point for 13:30 local solar time. The box shows the region where the classification is compared to that from Wernli et al. (2016).

Although the jet regime (J) is not excluded from the tropics by design, the RFO in this region is almost zero. It becomes more common in the southern ocean and also over some of the large scale descent regions, where the RFO of the frontal regime (which is assigned in preference to the jet-stream class) is much lower. The synoptic regimes are most common in the subtropical subsidence regions and the polar regions, where the RFO of the other regimes is small. However, there is also a significant RFO in other regions of the globe, demonstrating that even outside of the regions of large scale descent, it is still possible to find situations where cirrus cloud can form that is not clearly convective or frontal in origin.

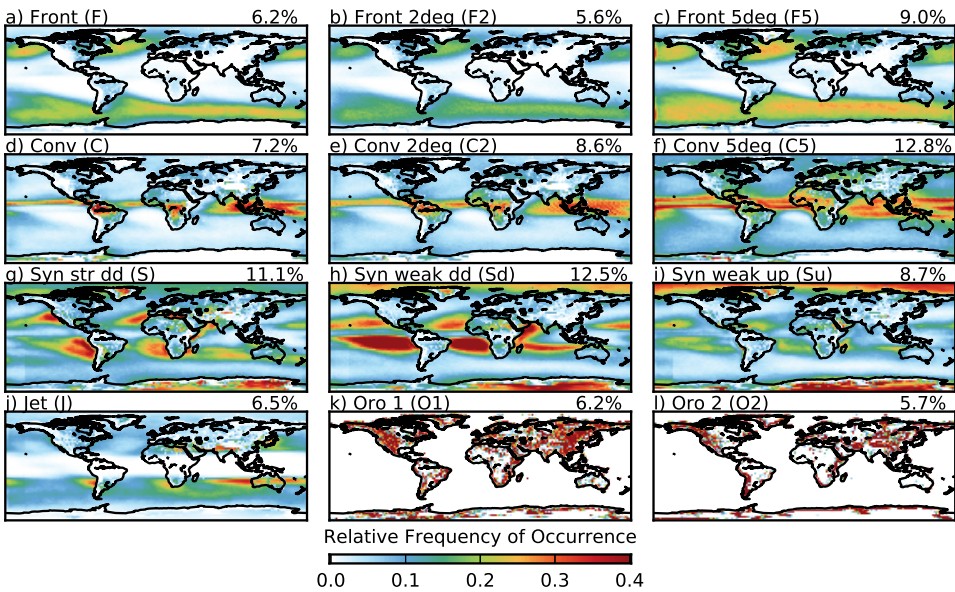

**Figure 2.** The relative frequency of occurrence of the different regimes over a ten year period (2003-2013). The frequencies are normalised so that they sum to one in each gridbox. The latitude weighted mean RFO for each regime is shown at the top right for each subplot.

## 3.3 Seasonal variation

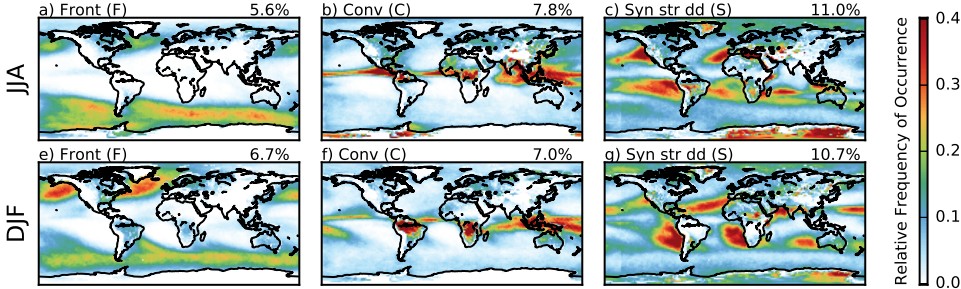

**Figure 3.** The relative frequency of occurrence for three of the regimes (frontal, convective and synoptic - strong downdraught) for the boreal summer (a,b,c) and winter (d,e,f) over a ten year period (2003-2013). The frequencies are normalised so that they sum to one in each gridbox. The latitude weighted mean RFO for each regime is shown at the top right for each subplot.

The global mean frequency of occurrence of the regimes is approximately constant between the seasons, but there is significant seasonal variability in the regime frequency of occurrence, particularly for the frontal, convective and synoptic regimes

(Fig. 3). Although it is concentrated in the mid-latitudes, the frontal regime is significantly more common in the winter hemisphere, with occurrences of around 10 to 20% in the north Pacific summer, but over 40% in during DJF. Over land, there is a slight increase in the occurrence of the frontal regime in the summer months, but this may be due a mis-classification of convective clouds in the frontal regime.

The variation in the convective regime in the tropics roughly follows the variation in precipitation (Adler et al., 2003, e.g.)with maxima in the Amazon and Congo regions during the boreal winter. There are also strong increases in the occurrence of the convective regime around south Asia in the summer, consistent with the occurrence of the monsoon. However, these increases are less clear over land, where the mountainous terrain often leads to a an orographic classification (Fig. 2k,ll). A winter increase in the convective regime is also observed over the Weddell sea, perhaps due to vertical motion generated by the mountains of the Antarctic Peninsular.

As a residual regime, the synoptic regime also shows seasonal variations in occurrence. However, these variations are less clearly a function of the meteorological situation. Variations in the synoptic regime occurrence follow those of the frontal regime in the mid-latitudes, with a higher synoptic regime occurrence in the summer hemisphere over the sub-tropical subsidence regions (Fig. 3c,f). These seasonal variations demonstrate the important of a regime decomposition when considering annual mean datasets.

## 3.4 Regime origins

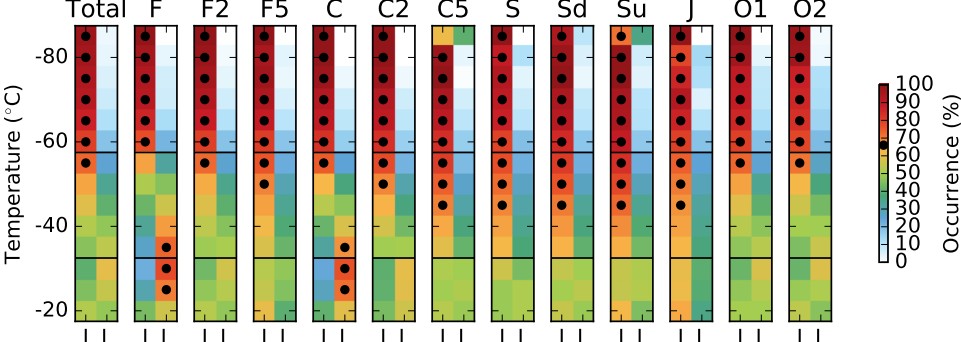

**Figure 4.** The probability of finding liquid or ice origin cirrus for each of the regimes over a north Atlantic region during 2007. The cirrus origin is from the dataset in Wernli et al. (2016), with "I" indicating "in-situ" origin and "L" indicating liquid origin. The plots are normalised such that the probabilities sum to one for each regime and temperature. The dots indicate more than two-thirds of clouds assigned to either liquid or in-situ origin classes.

Although these regimes have not been created using explicit information about their origin (does the ice originate from liquid droplets, or was it formed directly), they can be compared with the classification of Wernli et al. (2016) over the north Atlantic

to examine how skillfully they can determine the origin of different cirrus types. Fig. 4 shows the fraction of liquid and in-situ origin trajectories at each temperature level for each of the identified regimes during 2016 for the region depicted in Fig. 1c.

In all the regimes, almost all clouds colder than -60°C are formed directly as ice and many of those warmer than -40°C are originally formed as liquid (Fig. 4, "Total" column). However, there is considerable variation between the regimes between these temperatures.

The synoptic regimes (S, Sd, Su) are composed of mostly in-situ origin cirrus, even at relatively warm temperatures (close to -30°C), providing evidence that the cirrus clouds in these synoptic regimes are really formed directly in the ice phase. In contrast, the frontal and convective regimes are much more commonly liquid origin cirrus at temperatures warmer than -40°C, and even at temperatures as cold as -50°C, the proportions of liquid and in-situ origin cirrus are almost equal. The frontal regimes show a slightly higher proportion of liquid origin cloud around -45°C than the convective regimes. This may be related to reduced effectiveness of the back-trajectories in regions where the parametrised convection is responsible for much of the vertical transport. The two and five degree buffer regions for both the frontal (F2, F5) and convective regimes (C2, C5) show more similarities to the synoptic regimes than the main regimes (C, F), suggesting that 5 degrees is a suitable buffering distance.

## 3.5  Updraught velocities

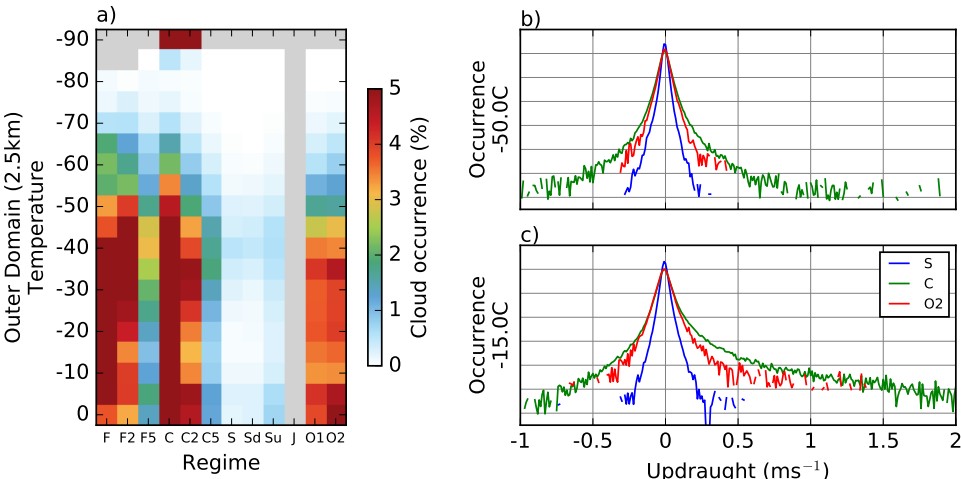

**Figure 5.** The properties of the observed regimes from within the one month ICON simulation, run in forecast mode. a) shows the vertical cloud occurrence fraction for each of the regimes. b) and c) show the updraught distributions for the synoptic, convective and orographic regimes at -50C (b) and -15°C (c). The grey lines in b) and c) are gridlines.

The convection-permitting ICON simulations for the tropical Atlantic in August 2016 show many of the expected properties of the regimes. The cloud occurrence (Fig. 5a) is highest for the frontal and convective regimes, becoming lower for the

buffer and the synoptic regimes. The orographic regimes also show an increased cloud fraction, although with lower cloud tops than the frontal and convective regimes. This demonstrates that the ICON simulation is able to adequately represent the meteorological situation, even when running in forecast mode and so can provide useful information on the properties of these regimes in the tropics.

There is a large variation in updraught velocity in the regimes (Figs. 5b,c), although this updraught variation is much more pronounced in the convective regime. The orographic regimes has a higher variability than the synoptic regime, but slightly lower than the convective regime, consistent with previous studies that have found high updraughts in orographic clouds. The convective regime has a long tail towards higher updraughts that is especially visible at lower levels in the atmosphere (Figs. 5b). This long tail results in the convective regime having a slightly larger mean in-cloud updraught than the synoptic regime (not

shown). The difference in mean updraught velocity is minimised as clouds are rarer in the synoptic regimes, forming only at the highest available updraughts within the regime.

At higher altitudes, the updraught distributions become more symmetrical. While the variability of the distributions (especially in the convective regime) is reduced, the convective and orographic regimes still have a broader distribution than the synoptic regime. This shows that even at high altitudes, higher updraughts are still more common in the convective and oro-

graphic regimes than in the synoptic regime. The results demonstrate that the classification proposed here is able to provide useful information on the vertical velocity environment of the clouds that cannot be resolved using reanalysis data.

### 3.6   Cloud optical depth by regime

There is not a strong distinction between cloud optical depths of most of the regimes. The ice optical depth frequency density is highest below an optical depth of 1 for the majority of the regimes. The main exceptions to this are the frontal (F) and

convective (C) regimes, as these regimes are selected based on their mean ice cloud optical depth. These regimes also have a much higher occurrence of ice cloud than the other regimes with fractional occurrences of 65 and 60% respectively. While the 2 degree buffer regimes C2 and F2 are noticeably different from the synoptic regimes, the 5 degree regimes are have a very similar optical depth distribution, again suggesting that 5 degrees is a suitable buffer distance. The orographic regimes both show a larger average cloud optical depth, particularly in the O2 regime, suggesting that the increased in-cloud updraught in

these and the convective regime (Fig. 5) has an important part to play in determining the cloud optical depth.

The liquid optical depths for the regimes are very similar between the regimes, with a maximum frequency density at an optical depth of about 5 for most regimes. Again, the F and C regimes have an optical depth distribution skewed towards larger values, but they have a lower fractional occurrence due to the overlying ice cloud. The similarities in the optical depth distributions of the regimes, despite the different updraught (Fig. 5) and ice origins (Fig. 4) of the regimes demonstrates how

the retrieved cloud properties alone are insufficient for fully identifying the cirrus cloud formation mechanisms.

### 3.7   Cloud radiative effect by regime

The CRE for each of the regimes (Fig. 7a) makes it clear that although some of the regimes have similar properties and origins, the mean CRE of the regimes occur in different locations in CRE space. The 25 and 75% quantiles for each of the regimes

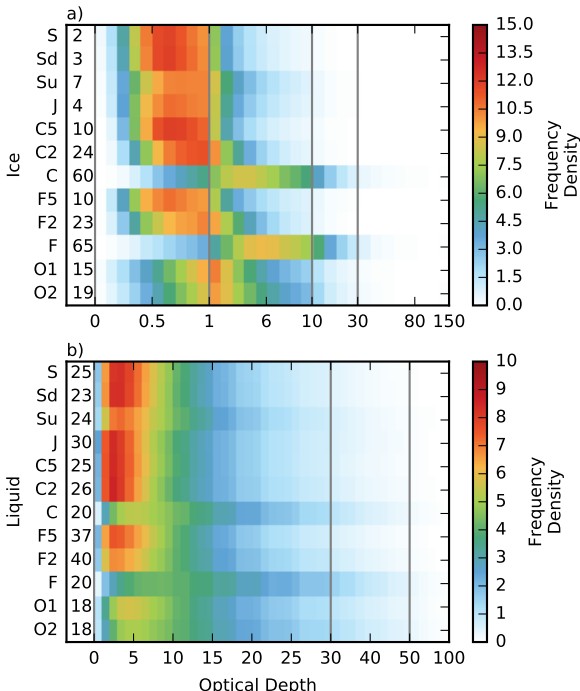

**Figure 6.** The mean normalised MODIS collection 6 cloud optical depth (COD) distribution for each of the regimes, showing clouds classified as a) ice-phase and b) liquid-phase separately. The bins are taken from the MODIS level 3 daily product, vertical lines indicate a change in bin width. The frequency density is normalised by the bin width and sums to one within each regime. The numbers along the right-hand edge of each plot show the percentage occurrence of valid retrievals within that regime (analogous to a cloud fraction). Note the different scale for the two plots.

indicate that the regimes are not as distinct radiatively as regimes defined using the cloud optical depth and CTP (Oreopoulos et al., 2016), with significant variation in CRE within the regimes.

Both the frontal and convective regimes have a strong negative SW and positive LW CRE, with the convective regime having a stronger LW and SW CRE than the frontal regime, although they both have very similar net CREs. This is due to the large optical depth of these regimes (Fig. 6). The two degree buffer regimes (C2, F2) fall between the main convective and frontal regimes and the rest of the cloud regimes, showing that the buffering is necessary to separate the synoptic regimes. The remainder of the regimes, including the five degree buffer regimes (C5, F5), have very similar CREs, suggesting that a five degree buffer region is sufficient to separate the synoptic regimes from the frontal and convective regimes. The jet stream cirrus regime (J) falls between F5 and the synoptic regimes, highlighting its composition as a hybrid between the frontal and the synoptic types. The synoptic regimes (S, Su and Sd) occupy a separate cluster, with the synoptic regimes having a smaller LW CRE, indicating a differing albedo-cloud top temperature relationship for these cloud types.

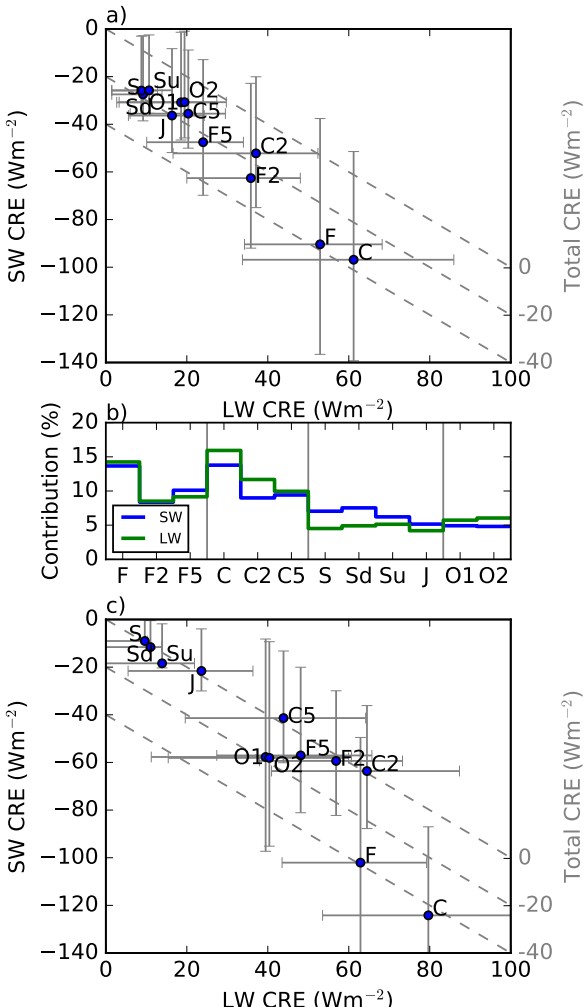

**Figure 7.** a) The daily, constant-meteorology mean cloud radiative effect (CRE) in the SW and LW for each of the regimes from CERES SYN1deg daily data. A negative value shows a cooling effect and the dashed grey lines are lines to constant net CRE. b) The contribution to the SW and LW global CRE from each regime. c) As (a), but using only pixels where 99% of the MODIS cloud top pressure retrievals are less than 550 hPa. The errorbars show the 25% and 75% quantiles for each regime.

Despite the strong CRE in the frontal and convective regimes, they do not dominate the global CRE in the same way (Fig. 7b), particularly in the SW, due to their low RFO of around 6% (Fig. 2). The frontal and convective regimes contribute around 15% of the global mean CRE in both the SW and the LW, while the synoptic and orographic regimes both contribute between 5% and 10%. However, it should be noted that this CRE is calculated for all of the clouds that occur in the gridbox, not just for the high clouds. As such, the occurrence of low clouds may influence the CRE calculated for the regimes, especially where the

high cloud is thin. Given the high RFO of the synoptic regimes in the subtropics, their large SW CRE contribution may be due to underlying stratocumulus clouds with a significant liquid cloud optical depth (Fig. 6b).

The contribution of the underlying clouds can be seen by comparing the results using only gridboxes with more than 99% of the MODIS observed cloud top pressures less than 550 hPa (Fig. 7c). While this does not completely separate the CRE of the high clouds (especially for vertically extensive clouds), it provides an idea of the CRE of the overlying high clouds. This indicates that the majority of the regimes have a very small contribution to the net CRE, with the SW and LW components roughly offsetting each other. The convective and frontal regimes are the exception, both having a strong negative total CRE.

One of the biggest changes is found in the synoptic regimes, where removing the low cloud has very little effect on the LW CRE, but a large reduction to the magnitude of the SW CRE, resulting in a slightly positive net CRE, similar to the results in previous work (Hartmann et al., 1992; Chen et al., 2000, e.g.)). This highlights the impact of the underlying low clouds in this regime and smaller effect of fractional cloud cover increases in the synoptic regime on the SW CRE due to this underlying low cloud. For the buffer regimes, the main effect of removing lower cloud is to increase the LW CRE. This is expected as it reduces the regime mean cloud top temperature, increasing the LW CRE. The small corresponding change in the SW CRE indicates that these regimes are not strongly affected by the presence of large amounts of low cloud.

## 4    Discussion

The results of the previous section show that by separating cirrus clouds according to their source, this classification provides information on the origin of the ice crystals in a cloud as well as the cloud-scale updraughts. This combination of satellite and reanalysis data provides extra information about the cloud properties that can be used to separate out cloud types and for future studies into cloud processes. However, there are still a number of improvements that could be made in future versions of the scheme.

One issue with this classification is that it is resolution dependent. This has a potential impact for defining the orographic regimes, where the windspeed-surface topography variation product used to define these regimes becomes small once the topography is sufficiently resolved. Defining the regimes at a different spatial resolution would require a re-calibration of some of the constants used in the classification. Whilst this could potentially be an issue for very high resolution models, the output from these models could be used on a lower resolution grid (as is demonstrated in section 3.5 of this work).

Another area for improvement is the use of the MODIS satellite data "blobs" within the scheme. These "blobs" are used to define regions of cloud that are connected and that intersect the meteorological conditions necessary for the formation of the frontal and convective types of cirrus, but this use highlights our current uncertainty about the best way to assign clouds to these regimes. These "blobs" require subjective choices for the definition of a "blob" and limit the regimes to being determined at 13:30 LST (the satellite overpass time). Future improvements to this scheme could use cloudy regions determined from the reanalysis, which would allow the classification to be generated at night, although the subjective definitions of cloud "blobs" would remain.

Another significant issue is that the classification is instantaneous, in that it only takes into account the meteorological and retrieved cloud properties at the moment of classification. As the classification has been designed for use with data from the A-train, this does not present a problem for investigating the properties of these regimes, but cirrus clouds are known to travel a considerable distance from convective source regions (Luo and Rossow, 2004). Although the convective regimes does a good

job of selecting high updraught clouds, there is clear scope for improvement. Possible methods include back-trajectories (e.g. Gehlot and Quaas, 2012) and cloud-object tracking, but until more information is available about the factors controlling cirrus cloud lifetime, conclusively linking an observed cirrus cloud to a convective event that took place days ago using reanalysis data continues to prove challenging. As many convective origin cirrus remain in the tropical region where the convective regime occurs, it is possible that they are already assigned to the convective (C) or buffer regimes (C2, C5), although future work will

be undertaken to explore this possibility.

## 5   Conclusions

In this work, a method of classifying cirrus clouds based on their origin has been demonstrated. This method makes use of re-analysis data to determine likely locations for frontal and convective cirrus. Combining this with satellite cloud observations from MODIS, cirrus clouds are assigned to frontal or convective regimes over ocean or regimes based on the surface roughness

over land. Any pixels that are not assigned to one of these classes are considered likely candidates for synoptic cirrus formation. The classification finds frontal regimes primarily in the extratropical storm-track regions (Fig. 2), with the convective regime occurring primarily in the tropics. Possible locations for synoptic clouds are found globally, particularly regions of large scale subsidence in the subtropics and polar regions. Orographic clouds are defined using a combination of the reanalysis windspeed and local sub-grid topography, similar to the parametrisations used in many GCMs. Significant seasonal variation is also

observed in the occurrence of the frontal, convective and synoptic regimes (Fig. 3).

When compared to the classification from (Wernli et al., 2016, Fig.4), it is shown that the regimes presented here are able to provide useful information on the origin of cirrus clouds (liquid or ice). The synoptic regimes in this classification are primarily composed of in-situ origin cirrus clouds, even to temperatures as warm as -20°C, while the frontal and convective regimes contain a much higher proportion of liquid-origin cirrus to much colder temperatures. At temperatures below -60°C,

almost all the observed clouds are in-situ origin cirrus.

Simulations with a high-resolution model show that the classification is also able to provide information on the updraught environment experienced by the clouds in each regime. In high-resolution simulations of the tropics, the convective regime has a significantly more variable updraught environment, with much more common strong updraughts and downdraughts than the synoptic regimes (Fig. 5). The convective regime also has a long tail of positive updraughts, leading to a higher mean in-cloud

updraught than found in the synoptic regimes. These results demonstrate the ability of this classification to provide information on the ice origin and in-cloud updraught that are not easily obtained from re-analysis data.

As seen in previous studies (e.g. Hartmann et al., 1992), the mean net cloud radiative effect (CRE) is negative, but with significant variation amongst the regimes (Fig. 7a). The frontal and convective regimes have the strongest LW, SW and net

negative CRE. The CRE for the synoptic regimes is strongly affected by underlying low cloud. Although the regime has a negative net CRE overall, when low clouds are removed, the net CRE is slightly positive due to a large reduction in the SW CRE (Fig. 7c). When regions with cloud top pressures lower than 550 hPa are removed, the net CRE for all of the regimes other than the frontal and convective regimes is close to zero.

Although there are some shortcomings to this classification, future work is planned to improve the selectivity and specificity of the regimes. However, they currently show significant skill in separating different cirrus types and provide a suitable starting point for investigating the differences between the properties and lifecycle of different cirrus types.

*Data access:* Preliminary agreement has been received to place the regime classification online at the British Atmospheric Data Centre (BADC) with a doi being assigned once the classification is finalised.

*Acknowledgements.* The authors would like to thank Heini Wernli (ETH Zürich) for providing data and for his helpful comments on the manuscript. The MODIS data are from the NASA Goddard Space Flight Center. The CERES data are from the NASA Langley Research Center. The GMTED2010 data is available from the U.S. Geological Survey. This work was supported by funding from the European Research Council under the European Union's Seventh Framework Programme (FP7/2007-2013) / ERC grant agreement no. FP7-306284 (QUAERERE). EG was supported by an Imperial College Junior Research Fellowship. TG received funding from the European Union's

Horizon 2020 research and innovation programme under the Marie Skłodowska-Curie Grant Agreement No. 703880. DK is supported by the Hans Ertel Center for Weather Research (HErZ), a German research network of universities, research institutions and the Deutscher Wetterdienst is funded by the BMVI (Federal Ministry of Transport and Digital Infrastructure).

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
