# Peer review of "An automated cirrus classification"

_Atmospheric Chemistry and Physics, 2017_

## Referee Comment (RC1) · Anonymous Referee #1 · 24 Oct 2017

Review of

**‚Technical note: An automated cirrus classification'**

by Gryspeerdt et al.

**General:**

In the manuscript,  a classification system for cirrus clouds that is based on re-analysis and satellite data is presented.  Cirrus clouds are separated in four main types, differing by meteorological/dynamical situation and thus microphysical and radiative properties. The topic of the study is very interesting and timely and I recommend the paper for publishing in ACP.

However, before final publication, I think that the manuscript should be revised taking into account the following points.

**1)** To my opinion, the study has more potential and relevance than currently elaborated. Though it is claimed to be a ‚technical note',  the link or physical mechanisms, respectively, between cirrus classes (meteorology), updraft, microphysical property (IWC or OD) and  radiative property (CRE) needs to be shown and discussed in more detail to make the study  scientifically sound.

The exciting is that with the applied method it seems that these links can be identified !

- In Fig. 4 the link between cirrus class and updraft is seen (the standard deviation of the respective updraft distribution could  serve as measure for class specific updraft);

- also, the mean CREs of the cirrus classes shown in Fig. 5 must be caused by a respective microphysical property (IWC, OD).

**2)** The aim of the paper is to  identify cirrus clouds by their formation mechanisms:
    orographic, frontal, convective,  in-situ

**2 a)**  The name ‚in-situ' does not match to the other names, which describe the meteorological situation  – it should be renamed to ‚synoptic'.

**2 b)** The defined classes refers to meteorological (dynamical) situations, not to formation mechanisms (as  stated in the abstract and elsewhere). Formation mechanisms are

–  homogeneous or heterogeneous ice nucleation  for in-situ origin cirrus (here called ice origin cirrus, see comment 3 below) and

–  heterogeneous or (sometimes) homogeneous drop freezing  for liquid origin cirrus.

So it should better be stated that cirrus clouds should be identified by the meteorological (dynamical) situation, which is what has been done in the paper.

**2 c)** I also recommend to link  the meteorological  to the dynamical  situation:

synoptic (in-situ), frontal, orographic and   convective cirrus    are cirrus in  increasing updraft regimes from low to high.

To identify cirrus by their formation mechanism, I would recommend to define for example three updraft regimes (weak, middle, high) and assign the them to the  meteorological types:

synoptic (in-situ)        - weak updraft,
frontal                 - middle updraft,
orographic/convective  - high updraft.

Then, the cirrus formation mechanisms can be identified (to a certain degree) by the updrafts:
weak updraft:         mostly heterogeneous freezing          – low IWC/OD    – low CRE ?
Middle/high updrafts:  increasing homogeneous ice formation – higher IWC/OD – higher  CRE ?

This is true for ‚liquid origin' as well as for ‚ice origin' cirrus.

As far as I can see, these links apparant in the paper and I would recommend to point that out in the paper.

**3)** ‚in-situ' : beside the previous comment on the term ‚in-situ' ( 2 a ), I also like to mention that ‚in-situ origin cirrus' is recently introduced (by Kraemer et al. (2016), ACP,  Luebke et al. (2016) and Wernli et al (2016), GRL) for  those cirrus that you name ‚ice origin cirrus'. Though ‚ice origin' might be the better companion of ‚liquid origin',  for consistency reasons I would recommend to keep the terms as they are  now introduced.

**4)** Cirrus formation mechanisms of in-situ origin and liquid origin cirrus and their link to cirrus properties and meteorological situations are also  discussed in Kraemer et al. (2016), ACP,  and Luebke et al. (2016), ACP.
Also, cirrus clouds classification, formation and so on is summarized in the recent review article of Heymsfield et al. (2017), Meteorological Monographs
(see  http://journals.ametsoc.org/doi/pdf/10.1175/AMSMONOGRAPHS-D-16-0010.1 ).

These studies should be considered in your work.

In more detail, Luebke et al. (2016) compared aircraft measurements in mid-latitude frontal liquid origin and in in-situ origin cirrus (+). They show the microphysical properties of the cirrus types and their distribution with temperature – which is quite similar to what is found in this study. This should be discussed, it is a good confirmation of  the approach used here.
(+) Liquid and in-situ cirrus are classified by means of trajectory analysis, similar as in Wernli et al. (2016).

**5)** The CRE is shown for the various cirrus types in Fig. 5 c). The highest total CRE is for F and C, followed by O1 and O2, and around zero CRE is for the other types. This seems to be related to the optical thickness or IWC, respectively of the cirrus types, which in turn depend on the updraft. A plot showing this would greatly improve the paper.

Also, it would be good to know if the cooling effect from F and C is because thick liquid origin cirrus constitute  a large part  of these cirrus types ?

In general, it would be good to see the difference in microphysical and radiative properties between liquid origin and ice origin in more detail.

**6)**  Methods:  The ‚Criteria for regime assignment' (please specify regime in Table 1, I guess the cirrus classes are meant)   are not very clear. I strongly recommend to add two columns, one containing the range of updrafts for each class and one with their range of microphysical properties, IWC or optical depth.

**7)** Abstract:  Include not only the method but also the most important results! In the current form, the paper will not  get much attention when potential readers look at the abstract – which I think is a pity.

**8)** Conclusions:  The properties of the cirrus classes are described, but I miss explanations of physical mechanisms leading to the properties. Two examples:

– ‚*The in-situ (*synoptic*) regimes in this classification are primarily composed of in-situ/ice-origin cirrus clouds, even to temperatures as warm as -20◦ C, while the frontal and convective regimes contain a much higher proportion of liquid-origin cirrus to much colder temperatures.'*

This is related to the updrafts, yes ? The larger the updrafts, the higher the liquid origin cirrus can rise = colder temperatures.

– ‚*The frontal and convective regimes have the strongest LW, SW and net negative CRE. ‚...'*

This could again be related to the updrafts, yes ?   High updrafts → thick cirrus, many liquid origin→ strong CRE, yes ?

This comment relates to comment 1) and 2) .

**Specific:**

**S1:** Page 1, line 15:  Please delete ‚*While'*

**S2:** Page 1, line 20-21: ‚*… aerosol influence on ice clouds would likely modify ice nucleation processes, changing the ICNC, perhaps by orders of magnitude …* '

‚orders of magnitude' is definitively too high, please scale back this statement. Also, aren't more recent publications available studying the effect of IN on cirrus properties ?

Another point to think about is that the most prominent parameter influencing the radiative cirrus properties is the ice water content (IWC). Changing the ICNC by influencing the IN number does not necessarily means that the IWC is changed, since the available water vapor distributes on the present ICNC. The result are different sizes of the ICNC (but not IWC) and thus differing sedimenation behavior, which influences the further development of the cloud.

**S3:** Page 1-2, lines 23-1:   ‚*... ice crystals are formed either by heterogeneous nucleation from ice nucleating particles (INP) or freezing of liquid droplets by either INP or existing ice crystals.*'

Do you mean either immersion freezing or contact freezing ? Please specify.

**S4:** Page 2, line 2:  ‚*... freezing of and remaining liquid droplets.*'   Please remove ‚and'.

**S5:** Page 2, line 5:     ‚*.. (e.g. Kärcher, 2017), ..*'    Since this is the introductary part of the manuscript, I would recommend to cite some more basic studies on the influence of freezing mechanisms on cirrus microphysical properties, e.g. the work of P. Spichtinger, E. Jensen, M. Kraemer, A. Heymsfield.
Include references.-→ Heymsfield 2017, review article.

**S6:** Page 2, line 6-7:   Heterogenous freezing in cirrus is in most cases determined by the INP number. This should be mentioned here.

**S7:** Page 2, line 8: ‚*Convective clouds* can *contain liquid water to temperatures as low as -37 ◦ C ...*'

**S8:** This happens only in very strong updrafts, please explain.

**S9:** Page 2, line 10:  ‚*... importance of the origin of the ice in a cloud (liquid or ice) has recently been* introduced and *demonstrated by Krämer et al. (2016).*'

**S10:** Page 2, lines 14-16: ‚*However, information on the in-cloud updraught and the ice origin has a strong dependence on the microphysics and convection schemes used in a model and so may not be suitable for use as an observations-based constraint on cloud ice microphysics parametrisations in general circulation models (GCMs).*'

To me this sentence is not very clear – can you reformulate what you mean ?

**S11:** Page 2: ‚***Existing classifications***'   I highly recommend to cite here the recent overview article of Heymsfield et al. (2017) (see h[ttp://journals.ametsoc.org/doi/pdf/10.1175/AMSMONOGRAPHS-D-16-0010.1](http://journals.ametsoc.org/doi/pdf/10.1175/AMSMONOGRAPHS-D-16-0010.1) ).

**S12:** Page 2, last paragraph:  This paragraph reads clumsy ….

**S13:** Page 3, lines 23-24: ,…, *irrespective of whether a cloud is observed such that a simpler comparison with models (which may produce sub-visible cirrus) can be made.*'    ????

**S14:** Page 4, lines 1-3:  What is the meaning of the ,windspeed-height variation product' that defines  O1 and O2?

**S15:**  Page 9, lines 6-8: ,*In all the regimes, almost all clouds colder than -60° C are formed directly as ice and many of those warmer than -40°C are originally formed as liquid (Fig. 3, "Total" column). However, there is considerable variation between the regimes between these temperatures.*'

This nice result should appear in the conclusions and maybe also in the abstract.

**S16 :** Page 14, lines 6-7: ,*The in-situ regimes in this classification are primarily composed of in-situ/ice-origin cirrus clouds, ...*'
     I guess you mean liquid here.

**S17 :** Page 14, line 16:  ,*As seen in previous studies, the net cloud radiative forcing (CRE) is negative, ...*'
      Which previous studies ?

---

## Referee Comment (RC2) · Anonymous Referee #2 · 3 Dec 2017

General Comments:

This is a very innovative approach for classifying the various types of cirrus clouds in a way that provides qualitative knowledge about cloud updrafts and whether the cirrus ice crystals formed near -38 °C from supercooled liquid cloud droplets advected into the T < -38 °C zone, classified as "liquid origin cirrus", or from another ice nucleation process (e.g. immersion freezing, vapor deposition or homogeneous freezing), classified as "ice origin cirrus". As such it represents a potentially significant contribution to scientific progress within the scope of ACP.

However, a key parameter not mentioned in the methodology for determining cirrus cloud regimes is temperature T. To begin with, the authors need to define what they mean by "cirrus cloud". Most investigators define cirrus as a pure ice cloud (i.e. no

liquid water is present), and the best way to insure this is to require T < -38 °C. Such a restriction was not applied in this study, making the proposed classification scheme ambiguous, especially in regards to cloud radiative properties. Unless I have not understood this classification scheme properly, this appears to be the main drawback.

It is evident from Fig. 3 that the classification scheme is applied for T < -20 °C, and supercooled liquid water may exist between -20 and -38 °C. Over this T range, the clouds should not be regarded as cirrus clouds. The differences between the cirrus categories in Fig. 3 become much more subtle if cirrus are defined as being colder than -40 °C, but the cloud categories can be distinct for -40 °C < T < -20 °C. Perhaps this classification scheme could be improved if each class were divided into two T regimes; T < -40 °C and -40 °C < T < -20 °C.

Figure 4 introduces even more mixed phase ambiguity by applying the classification scheme to cloud temperatures between 0 °C and -90 °C.

Although I am familiar with the concepts of liquid origin and ice origin cirrus clouds, I felt that these complex concepts were not clearly explained in this paper, especially in regards to what kind of knowledge they impart to this classification scheme. More explanation should be given.

The paper is well organized and well written, with a sufficient number of quality figures to illustrate the main points. Many other important concerns are listed below. Given these concerns, I recommend major revisions.

Major Comments:

1. Page 2, line 18: In this section on "Existing Classifications", the authors might also want to mention the work of Tselioudis et al. (2013, J. Climate), who used cluster analysis to define 11 atmospheric weather states (WSs) based on optical depth and pressure level. While only one WS is primarily cirrus, other WSs contain cirrus contributions.

2. Page 3, line 15: This is the 1st mention of the ICON model that is used extensively in this work. The full name of the model and/or a reference should be given here (along with acronym).

3. Page 3, line 31: In some mountainous regions, 850 hPa may be below the surface. What is done when this occurs?

4. Section 2 (Methods): Since MODIS was used to develop this classification scheme, it would be helpful to show in this paper a mean visible cloud optical depth (OD) associated with each cirrus cloud category, as well as the corresponding standard deviations. This would be helpful for understanding the net CRE of each category that is discussed later.

5. Section 2 (Methods): A temperature criteria of T < -38 °C is not used to select cirrus in any of these cirrus categories, raising the possibility that some clouds classified as cirrus may actually be mixed phase clouds. Figures 3 and 4 suggest that this classification scheme was applied for T < -20 °C and T < 0 °C, respectively. If either is correct, then mixed phase conditions are built into this classification scheme, and this should be made clear. Moreover, the word "cirrus" in the paper's title should be replaced by "ice cloud", and all references to "cirrus" in the paper should be replaced by "ice cloud".

6. Page 9, lines 2-8: Since this classification scheme is for cirrus clouds, this implies only ice exists. But when classifying clouds between -20 and -40 C, what assurance is there that these clouds are "ice only" based on Wernli et al. (2016)? And even if the Wernli et al. analysis shows that the classified "cirrus" in Fig. 3 between -20 and -40 C are ice only, the phase partitioning in cloud resolving models is not an exact science, is highly variable between models, and depends strongly on the parameterization scheme used. Thus it is difficult to understand just what exactly is being shown in Fig. 3 at warmer temperatures (e.g. are the clouds ice only or mixed phase?). Please address these concerns, clarifying all these issues. If the authors insist on using their classification scheme at these warmer temperatures, they need to

be clear just what kind of cloud they are classifying (e.g. all-ice or mixed phase).

7. Figure 5c and associated discussion: Two questions come to mind here: (1) How much do mixed phase conditions contribute to these CRE values? Even if liquid water comprises only 10% of the total water content, it can still have a large impact on cloud radiative properties (e.g. Mitchell and d'Entremont, 2012, AMT; Shupe and Intrieri, 2004, J. Climate). Thus, a small liquid water fraction is likely to have a strong impact on the net CREs given in Fig. 5c, increasing the SW over the LW contribution.

8. And regarding the 2nd question (wrt Fig. 5c), CRE is evaluated from CERES SYN1deg daily data at 1:30 pm LST. At this time, SW CRE is near maximum, whereas LW CRE is much less variable over a 24 hr. daily cycle. This sampling time will negatively bias the net CRE, making it non-representative of the daily-mean net CRE associated with cirrus clouds having low-to-moderate ODs.

9. Fig. 5b and c: It is commendable that the authors have partly explained why all the in situ cirrus categories have more SW CRE than LW CRE (due to low clouds). These cirrus are typically having lower optical depth and thus lower SW & LW CRE (Fig. 5c), with TOA LW CRE > SW CRE (e.g. Fu, 2008, Fig. 4; Hong and Liu, 2015, J. Climate). But after removing the low clouds in Fig. 5c, in situ cirrus still have a net CRE $\sim$ zero, whereas other studies infer positive values. For example, for cirrus OD < 3.6 and cloud top pressure <440 mb, the net CRE reported by Chen et al. (2000, J. Clim.) was positive, as was also true for Hartman et al. (1992, J. Clim.) for cirrus OD < 9.4. Based on the ECHAM6 GCM, the global average net CRE of cirrus clouds is +5.7 W/m2 (Gasparini & Lohmann, 2016, JGR). The proposed technical note appears to be at variance with the literature in regards to the overall sign of the net cirrus forcing, and this discrepancy should be addressed. Note that the calculations in Fu are for the equator during an equinox when the sun is highest in the sky, which maximizes the SW CRE.

10. Page 14, line 16: "As seen in previous studies, the net cloud radiative forcing (CRE)

is negative". Yes, but this paper is about cirrus clouds, and their net CRE is positive. Please cite these "previous studies" that pertain to cirrus clouds. One study by Chen et al. (2000, J. Clim.) was cited in the Introduction and could be cited again here. As noted above, Chen et al. (2000) and Hartman et al. (1992) found that for cirrus OD < 3.6 or 9.4, respectively, their net CRE is positive.

Minor Comments:

1. Page 2, line 2: Remove "and" from this sentence.

2. Page 2, line 12: Comma not needed

3. Page 2, line 18: "Existing Classifications" should be given a sub-header value of 3.1.

4. Page 3, line 27: determines => determining?

5. Figure 4: Please label all the panels as a, b and c. Also, what do the 3 horizontal lines indicate in the middle-regions of Fig. 4b and 4c?

6. Page 12, line 15: Fig. 5a does not show RFO; please clarify. Also, "frontal convective regimes" => "frontal and convective regimes"? Based on Fig. 5b, frontal & convective regimes appear to account for slightly > 12%.

7. Page 13, line 8: "once" => "until"?

———————————————————

---

## Author Comment (AC1) · 5 Feb 2018

* * *
*: In the manuscript, a classification system for cirrus clouds that is based on re-analysis and satellite data is presented. Cirrus clouds are separated in four main types, differing by meteorological/dynamical situation and thus microphysical and radiative properties. The topic of the study is very interesting and timely and I recommend the paper for publishing in ACP. However, before final publication, I think that the manuscript should be revised taking into account the following points.*

**Reply**: We thank the reviewer for their useful comments and address each of them in turn below. Line numbers related to the diff'ed version of the manuscript. A short section on the seasonal variation of the regimes has been added to demonstrate their

utility and help better characterise them for future work.

*1: To my opinion, the study has more potential and relevance than currently elaborated. Though it is claimed to be a "technical note", the link or physical mechanisms, respectively, between cirrus classes (meteorology), updraft, microphysical property (IWC or OD) and radiative property (CRE) needs to be shown and discussed in more detail to make the study scientifically sound. The exciting is that with the applied method it seems that these links can be identified!*

- *In Fig. 4 the link between cirrus class and updraft is seen (the standard deviation of the respective updraft distribution could serve as measure for class specific updraft);*

- *Also, the mean CREs of the cirrus classes shown in Fig. 5 must be caused by a respective microphysical property (IWC, OD).*

**Reply**: This paper was originally pitched as a technical note, as it aimed primarily to describe the occurrence of the regimes and how they are defined. With the inclusion of the model output and CRE data it has become a bit more in-depth and so it has now been shifted so that it is listed as a research article. We have re-worded parts of the paper to make these links more explicit and have included information about the cloud optical depth in Fig. 6. However, We do not believe that we are yet able to account for the properties of the clouds within regimes. While this study shows that the convective and orographic regimes have a higher updraught than the synoptic regime, other properties, such as the aerosol environment remain uncertain, preventing an attribution of the differences in radiative and cloud properties to any particular factor. Further studies are planned to explore these differences in more detail.

*2a: The aim of the paper is to identify cirrus clouds by their formation mechanisms: orographic, frontal, convective, in-situ. The name "in-situ" does not match to the other names, which describe the meteorological situation - it should be renamed to "synoptic".*

**Reply**: Thank you for this comment, "synoptic" is a better fit for this class and it has now been renamed.
* * *
*2b: The defined classes refers to meteorological (dynamical) situations, not to formation mechanisms (as stated in the abstract and elsewhere). Formation mechanisms are:*

- *homogeneous or heterogeneous ice nucleation for in-situ origin cirrus (here called ice origin cirrus, see comment 3 below) and*

- *heterogeneous or (sometimes) homogeneous drop freezing for liquid origin cirrus.*

*So it should better be stated that cirrus clouds should be identified by the meteorological (dynamical) situation, which is what has been done in the paper.*

**Reply**: Thank you for pointing out the potential confusion here. Following previous regime definitions, it is not clear that "dynamical regimes" should be used either, as it has previously been applied to regimes defined using only meteorological variables (e.g. Nam and Quaas, 2013; Zhang et al., 2016). We note that some previous studies (e.g. Heymsfield et al., 2017) refer to ice formation mechanisms (homogeneous and heterogeneous) somewhat separately from the mechanisms that create the cloud in the first place (processes creating the cloud updraught, radiative cooling). We have modified the text to refer to cirrus or cloud formation mechanisms where there is a potential for misunderstanding.
* * *
*2c: I also recommend to link the meteorological to the dynamical situation: synoptic (in-situ), frontal, orographic and convective cirrus are cirrus in increasing updraft regimes from low to high. To identify cirrus by their formation mechanism, I would recommend to define for example three updraft regimes (weak, middle, high) and assign the them to the meteorological types:*

- *synoptic (in-situ) - weak updraft,*

- *frontal - middle updraft,*

- *orographic/convective - high updraft.*

*Then, the cirrus formation mechanisms can be identified (to a certain degree) by the updrafts*

- *weak updraft: mostly heterogeneous freezing - low IWC/OD - low CRE ?*

- *Middle/high updrafts: increasing homogeneous ice formation - higher IWC/OD - higher CRE ?*

*: This is true for "liquid origin" as well as for "ice origin" cirrus. As far as I can see, these links apparant in the paper and I would recommend to point that out in the paper.*
**Reply**: We have added some more discussion on the updraughts within the different regimes (e.g. P14L1), but we do not believe that there is enough information to add updraught information to the classification as it stands. The model results with ICON indicate that there is a link between the regime type and the cloud scale updraughts, but without more large scale measurement efforts, we do not believe it is possible to classify the regimes by updraught yet.
* * *
*3: "in-situ" : beside the previous comment on the term "in-situ" (2a), I also like to mention that "in-situ origin cirrus" is recently introduced (by Kraemer et al. (2016),*

*ACP, Luebke et al. (2016) and Wernli et al (2016), GRL) for those cirrus that you name "ice origin cirrus". Though "ice origin" might be the better companion of "liquid origin", for consistency reasons I would recommend to keep the terms as they are now introduced.*
**Reply**: references to "ice-origin" have now been updated to "in-situ origin" in line with previous work.
* * *
*4: Cirrus formation mechanisms of in-situ origin and liquid origin cirrus and their link to cirrus properties and meteorological situations are also discussed in Kraemer et al. (2016), ACP, and Luebke et al. (2016), ACP. Also, cirrus clouds classification, formation and so on is summarized in the recent review article of Heymsfield et al. (2017), Meteorological Monographs (see http://journals.ametsoc.org/doi/pdf/10.1175/AMSMONOGRAPHS-D-16-0010.1). These studies should be considered in your work. In more detail, Luebke et al. (2016) compared aircraft measurements in mid-latitude frontal liquid origin and in in-situ origin cirrus. They show the microphysical properties of the cirrus types and their distribution with temperature, which is quite similar to what is found in this study. This should be discussed, it is a good confirmation of the approach used here. Liquid and in-situ cirrus are classified by means of trajectory analysis, similar as in Wernli et al (2016)*
**Reply**: Thank you for bringing these studies to our attention, they are now included in the introduction.
* * *
*5: The CRE is shown for the various cirrus types in Fig. 5 c). The highest total CRE is for F and C, followed by O1 and O2, and around zero CRE is for the other types. This seems to be related to the optical thickness or IWC, respectively of the cirrus types, which in turn depend on the updraft. A plot showing this would greatly improve the paper. Also, it would be good to know if the cooling effect from F and C is because thick liquid origin cirrus constitute a large part of these cirrus types ? In general, it would be good to see the difference in microphysical and radiative properties between*

*liquid origin and ice origin in more detail.*

**Reply**: A plot showing the COD for the regimes has been included in Fig. 6. The large negative CRE of the F and C classes is indeed due to their large COD, as they are the only regimes that have selection criteria based on their COD (mentioned P14L14). While it would make sense, unfortunately we do not have enough data to know if the liquid origin cirrus in the F and C regimes is the dominant cause of the strong negative CRE, as the liquid-ice origin classification only exists for a single year over the North Atlantic.

*6: Methods: The "Criteria for regime assignment" (please specify regime in Table 1, I guess the cirrus classes are meant) are not very clear. I strongly recommend to add two columns, one containing the range of updrafts for each class and one with their range of microphysical properties, IWC or optical depth.*

**Reply**: Information on the optical depth is now included in Fig. 6. The table currently contains the information necessary to reproduce the regime classification. We would prefer not to add information on the cirrus class properties at this point in the paper, as they are not used in the regime classification and are now covered in more detail later in the paper (e.g. Fig. 6)

*7: Abstract: Include not only the method but also the most important results! In the current form, the paper will not get much attention when potential readers look at the abstract – which I think is a pity.*

**Reply**: Information on the results has now been included.

*8: Conclusions: The properties of the cirrus classes are described, but I miss explanations of physical mechanisms leading to the properties. Two examples:*

- *"The in-situ (synoptic) regimes in this classification are primarily composed of in-situ/ice-origin cirrus clouds, even to temperatures as warm as -20° C while the*

*frontal and convective regimes contain a much higher proportion of liquid-origin cirrus to much colder temperatures." This is related to the updrafts, yes? The larger the updrafts, the higher the liquid origin cirrus can rise = colder temperatures.*

- *"The frontal and convective regimes have the strongest LW, SW and net negative CRE." This could again be related to the updrafts, yes? High updrafts → thick cirrus, many liquid origin → strong CRE, yes?*

*This comment relates to comments 1 and 2.*
**Reply**: This may be the case, but we do not feel that we have enough data to claim that this is purely an updraught effect. For example, it could be that the cloud bases are higher in the synoptic class (as indicated by the model results), rather than just a change in the cloud top height of clouds with lower bases. A comment related to the updraught has been added to the (new) section on the cloud optical depth (P13L3).

Specific comments
* * *
**Page 1, line 15::** *Please delete "While"*
**Reply**: Done
* * *
**Page 1, line 20-21::** *"... aerosol influence on ice clouds would likely modify ice nucleation processes, changing the ICNC, perhaps by orders of magnitude ..." - "orders of magnitude" is definitively too high, please scale back this statement. Also, aren't more recent publications available studying the effect of IN on cirrus properties ? Another point to think about is that the most prominent parameter influencing the radiative cirrus properties is the ice water content (IWC). Changing the ICNC by influencing the IN number does not necessarily means that the IWC is changed, since the available water vapor distributes on the present ICNC. The result are different sizes of the ICNC*

*(but not IWC) and thus differing sedimenation behavior, which influences the further development of the cloud.*

**Reply**: This has been changed to "an order of magnitude", the maximum change observed in the Kärcher and Lohmann, (2003) paper. We agree that the IWC is a more important component of the cloud albedo and that a change in ICNC may not modify it. However, even at a constant IWC, a change in the ICNC would modify the radiative properties of the cloud, similar to the Twomey effect in liquid clouds. The IWC has been included in the first paragraph along with the ICNC (P1L21).
* * *
***Page 1-2, lines 23-1***: *"... ice crystals are formed either by heterogeneous nucleation from ice nucleating particles (INP) or freezing of liquid droplets by either INP or existing ice crystals." Do you mean either immersion freezing or contact freezing ? Please specify.*

**Reply**: This sentence was intended to provide a brief mention of the role of INP. It has been expanded to indicate that both contact and immersion freezing are possible.
* * *
***Page 2, line 2***: *"..freezing of and remaining liquid droplets." Please remove "and".*
**Reply**: Done
* * *
***Page 2, line 5***: *".. (e.g. Kärcher, 2017), .." Since this is the introductary part of the manuscript, I would recommend to cite some more basic studies on the influence of freezing mechanisms on cirrus microphysical properties, e.g. the work of P. Spichtinger, E. Jensen, M. Kraemer, A. Heymsfield. Include references.-→ Heymsfield 2017, review article.*

**Reply**: Thank you for pointing this out, we have now included the Heymsfield et al., 2017 review article here.
* * *
***Page 2, line 6-7***: *Heterogenous freezing in cirrus is in most cases determined by the INP number. This should be mentioned here.*

**Reply**: Done
* * *
**Page 2, line 8::** *"Convective clouds can contain liquid water to temperatures as low as -37°C ..." - This happens only in very strong updrafts, please explain.*
**Reply**: A clarification about the high updraught speeds is now included.
* * *
**Page 2, line 10::** *... importance of the origin of the ice in a cloud (liquid or ice) has recently been introduced and demonstrated by Krämer et al. (2016).*
**Reply**: Amended.
* * *
**Page 2, lines 14-16::** *"However, information on the in-cloud updraught and the ice origin has a strong dependence on the microphysics and convection schemes used in a model and so may not be suitable for use as an observations-based constraint on cloud ice microphysics parametrisations in general circulation models (GCMs)." - To me this sentence is not very clear - can you reformulate what you mean ?*
**Reply**: Replaced with "However, the cloud scale updraught and the ice origin are not often directly simulated in atmospheric models, being calculated through parametrisations. As such, reanalysis values of these quantities may not be suitable for use as a constraint on cloud ice microphysics parametrisations in general circulation models (GCMs)."
* * *
**Page 2::** *Existing classifications - I highly recommend to cite here the recent overview article of Heymsfield et al. (2017)(see http://journals.ametsoc.org/doi/ pdf/10.1175/AMSMONOGRAPHS-D-16-0010.1).*
**Reply**: This reference has now been included, although slightly earlier in the manuscript (P2L13) so that it provides a good reference for the introduction on cirrus in general.
* * *
**Page 2, last paragraph::** *This paragraph reads clumsy*

**Reply**: This paragraph has been re-worded.
* * *
***Page 3, lines 23-24::*** *"..., irrespective of whether a cloud is observed such that a simpler comparison with models (which may produce sub-visible cirrus) can be made."* *????*

**Reply**: This is simplified to "...irrespective of whether a cloud is observed. This ensures that every location is assigned to a regime, such that the regime occurrence is not biased by any satellite cloud detection threshold." We has also removed the "aim of classifying the uppermost layer" from later in this paragraph due to it's potential to confuse this issue.
* * *
***Page 4, lines 1-3::*** *What is the meaning of the "windspeed-height variation product" that defines O1 and O2?*

**Reply**: This has been changed to "windspeed-surface topography variation product" to better align with the terminology in the previous paragraph.
* * *
***Page 9, lines 6-8::*** *"In all the regimes, almost all clouds colder than -60$^\circ$C are formed directly as ice and many of those warmer than -40$^\circ$C are originally formed as liquid (Fig. 3, "Total" column). However, there is considerable variation between the regimes between these temperatures."* *This nice result should appear in the conclusions and maybe also in the abstract.*

**Reply**: The abstract has been re-worded to make a clearer link to the variations in ice origin between the regimes. The temperature dependence of the in-situ/liquid origin cirrus had previously been noted in the Wernli et al, 2017, paper describing the classification.
* * *
***Page 14, lines 6-7::*** *"The in-situ regimes in this classification are primarily composed of in-situ/ice-origin cirrus clouds, ..."* *I guess you mean liquid here.*

**Reply**: This sentence was intended to show that the liquid-origin cirrus is much less

common in the synoptic regime at warmer temperatures than in the frontal or convective regimes. The renaming of the in-situ regime to synoptic should make this clearer.

*Page 14, line 16::* *"As seen in previous studies, the net cloud radiative forcing (CRE) is negative, ..." - Which previous studies ?*

**Reply**: This was intended to reference the overall mean CRE. This sentence has now been corrected to read "As seen in previous studies (e.g. Hartmann et al., 1992), the mean cloud radiative effect (CRE) is negative ..."

**References**

Hartmann, D. L., Ockert-Bell, M. E., and Michelsen, M. L.: The Effect of Cloud Type on Earth's Energy Balance: Global Analysis, J. Climate, 5, 1281–1304, doi:10.1175/1520-0442(1992)005<1281:TEOCTO>2.0.CO;2, 1992.

Heymsfield, A. J., Krämer, M., Luebke, A., Brown, P., Cziczo, D. J., Franklin, C., Lawson, P., Lohmann, U., McFarquhar, G., Ulanowski, Z., and Van Tricht, K.: Cirrus Clouds, Meteorological Monographs, 58, 2.1–2.26, doi:10.1175/AMSMONOGRAPHS-D-16-0010.1, 2017.

Medeiros, B. and Stevens, B.: Revealing differences in GCM representations of low clouds, Climate Dyn., 36, 385–399, doi:10.1007/s00382-009-0694-5, 2011.

Nam, C. C. W. and Quaas, J.: Geographical versus dynamically defined boundary layer cloud regimes and their use to evaluate general circulation model cloud parameterizations, Geophys. Res. Lett., doi:10.1002/grl.50945, 2013.

Zhang, S., Wang, M., Ghan, S. J., Ding, A., Wang, H., Zhang, K., Neubauer, D., Lohmann, U., Ferrachat, S., Takeamura, T., Gettelman, A., Morrison, H., Lee, Y., Shindell, D. T., Partridge, D., Stier, P., Kipling, Z., and Fu, C.: On the characteristics of aerosol indirect effect based on dynamic regimes in global climate models, Atmos. Chem. Phys., 16, 2765–2783, doi:10.5194/acp-16-2765-2016, 2016.

---

## Author Comment (AC2) · 5 Feb 2018

General Comments:
* * *
*: This is a very innovative approach for classifying the various types of cirrus clouds in a way that provides qualitative knowledge about cloud updrafts and whether the cirrus ice crystals formed near -38°C from supercooled liquid cloud droplets advected into the T < -38°C zone, classified as "liquid origin cirrus", or from another ice nucleation process (e.g. immersion freezing, vapor deposition or homogeneous freezing), classified as "ice origin cirrus". As such it represents a potentially significant contribution to scientific progress within the scope of ACP.*

[Figure]

**Reply**: We thank the reviewer for their comments, which are addressed in turn below. Line numbers related to the diff'ed version of the manuscript. A short section on the seasonal variation of the regimes has been added to demonstrate their utility and help better characterise them for future work.

*: However, a key parameter not mentioned in the methodology for determining cirrus cloud regimes is temperature T. To begin with, the authors need to define what they mean by "cirrus cloud". Most investigators define cirrus as a pure ice cloud (i.e. no liquid water is present), and the best way to insure this is to require $T < -38°C$. Such a restriction was not applied in this study, making the proposed classification scheme ambiguous, especially in regards to cloud radiative properties. Unless I have not understood this classification scheme properly, this appears to be the main drawback.*
**Reply**: We agree that this is an important factor of cirrus clouds. This classification is primarily intended to classify gridboxes, irrespective of the cloud that had actually formed there, similar to the dynamic regimes described in Medeiros and Stevens (2011). As noted here, this then leads to the issue observed in Fig. 6, where the CRE is a strong function of the underlying low-level cloud. Although we have not completely removed cloud properties from the regime classification, having a classification that is mostly independent of the cloud properties then allows global models to be assessed on their ability to form these regimes separately from their simulation of the cloud properties within them. This has been expanded upon in the methods section (P4L11) and noted at the end of the introduction (P4L5).

*: It is evident from Fig. 3 that the classification scheme is applied for $T < -20°C$, and supercooled liquid water may exist between -20 and $-38°C$. Over this T range, the clouds should not be regarded as cirrus clouds. The differences between the cirrus categories in Fig. 3 become much more subtle if cirrus are defined as being colder than $-40°C$, but the cloud categories can be distinct for $-40°C < T < -20°C$. Perhaps this classification scheme could be improved if each class were divided into two T regimes; $T < -40°C$*

*and -40°C < T < -20°C.*

**Reply**: Following the previous point, classification according to the observed cloud phase would conflate the occurrence of the regimes with the occurrence of different cloud types within them. This also limits the possibility of splitting the regime by temperature vertically by cloud occurrence.

The liquid/in-situ origin dataset is derived from ERA-Interim, which contains a relatively simple ice parametrisation, lacking the impact of aerosols such that most clouds are glaciated by -20°C. This means that the occurrence of liquid-origin cirrus becomes very low even at temperatures warmer than -40°C, whilst this might not be the case in the atmosphere, where liquid drops can persist to colder temperatures (possibly allowing liquid origin cirrus a colder temperatures). This then suggests that the difference between the regimes might persist to higher altitudes in the atmosphere, even though it cannot be determined from this dataset. The methods section has been expanded to include this point (P7L8).
* * *
*: Figure 4 introduces even more mixed phase ambiguity by applying the classification scheme to cloud temperatures between 0°C and -90°C.*
**Reply**: See previous points
* * *
*: Although I am familiar with the concepts of liquid origin and ice origin cirrus clouds, I felt that these complex concepts were not clearly explained in this paper, especially in regards to what kind of knowledge they impart to this classification scheme. More explanation should be given.*
**Reply**: Extra explanation is included in the introduction (P2L19) and in the relevant results sections.

The paper is well organized and well written, with a sufficient number of quality figures to illustrate the main points. Many other important concerns are listed below. Given these concerns, I recommend major revisions.

[Figure]

Major Comments:

**Page 2, line 18::** *In this section on "Existing Classifications", the authors might also want to mention the work of Tselioudis et al. (2013, J. Climate), who used cluster analysis to define 11 atmospheric weather states (WSs) based on optical depth and pressure level. While only one WS is primarily cirrus, other WSs contain cirrus contributions.*
**Reply**: Thank you for suggesting this, it has now been included.

**Page 3, line 15::** *This is the 1st mention of the ICON model that is used extensively in this work. The full name of the model and/or a reference should be given here (along with acronym).*
**Reply**: The acronym and reference are now included here

**Page 3, line 31::** *In some mountainous regions, 850 hPa may be below the surface. What is done when this occurs?*
**Reply**: Following the ERA-Interim extrapolation algorithm, this windspeed is replaced with the surface windspeed. This is now noted in the text.

**Section 2 (Methods)::** *Since MODIS was used to develop this classification scheme, it would be helpful to show in this paper a mean visible cloud optical depth (OD) associated with each cirrus cloud category, as well as the corresponding standard deviations. This would be helpful for understanding the net CRE of each category that is discussed later.*
**Reply**: This is now included towards the end of the results section (Fig. 6)

**Section 2 (Methods)::** *A temperature criteria of $T < -38°C$ is not used to select cirrus in any of these cirrus categories, raising the possibility that some clouds classified as*

*cirrus may actually be mixed phase clouds. Figures 3 and 4 suggest that this classification scheme was applied for T < -20° C and T < 0° C, respectively. If either is correct, then mixed phase conditions are built into this classification scheme, and this should be made clear. Moreover, the word "cirrus" in the paper's title should be replaced by "ice cloud", and all references to "cirrus" in the paper should be replaced by "ice cloud".*
**Reply**: Thank you for pointing this out, we agree that the phase is ambiguous for some of these cloud, but as mentioned previously, we feel that adding cloud phase to the classification would reduce the separation between the meteorological state and the cloud formed due to it. While "ice cloud" or "cirriform" might be a broader term to cover this classification, we feel that the term "cirrus" is also a good way to indicate that this classification aims to separate out many different types of high cloud, from anvil cirrus through to the thinner synoptic cirrus varieties.
* * *
*Page 9, lines 2-8::* *Since this classification scheme is for cirrus clouds, this implies only ice exists. But when classifying clouds between -20 and -40 C, what assurance is there that these clouds are "ice only" based on Wernli et al. (2016)? And even if the Wernli et al. analysis shows that the classified "cirrus" in Fig. 3 between - 20 and -40 C are ice only, the phase partitioning in cloud resolving models is not an exact science, is highly variable between models, and depends strongly on the parameterization scheme used. Thus it is difficult to understand just what exactly is being shown in Fig. 3 at warmer temperatures (e.g. are the clouds ice only or mixed phase?). Please address these concerns, clarifying all these issues. If the authors insist on using their classification scheme at these warmer temperatures, they need to be clear just what kind of cloud they are classifying (e.g. all-ice or mixed phase).*
**Reply**: Following the points made earlier, we have noted that the classification attempts to avoid basing the classification on the retrieved cloud properties (such as phase), to enable a clearer distinction between the meteorological state and the clouds it produces. Further notes on this have been included in the methods section (P4L11) and introduction to better justify this.
* * *
*Figure 5c and associated discussion: : Two questions come to mind here: (1) How much do mixed phase conditions contribute to these CRE values? Even if liquid water comprises only 10% of the total water content, it can still have a large impact on cloud radiative properties (e.g. Mitchell and d'Entremont, 2012, AMT; Shupe and Intrieri, 2004, J. Climate). Thus, a small liquid water fraction is likely to have a strong impact on the net CREs given in Fig. 5c, increasing the SW over the LW contribution.*
**Reply**: It is likely that mixed-phase clouds play a role in the CRE of these regimes, especially given the importance of liquid water in these clouds. An investigation into this is beyond the scope of this work, which is mainly to provide a description and brief characterisation, but is definitely considered for the future.
* * *
*Figure 5c: And regarding the 2nd question (wrt Fig. 5c), CRE is evaluated from CERES SYN1deg daily data at 1:30 pm LST. At this time, SW CRE is near maximum, whereas LW CRE is much less variable over a 24 hr. daily cycle. This sampling time will negatively bias the net CRE, making it non-representative of the daily-mean net CRE associated with cirrus clouds having low-to-moderate ODs.*
**Reply**: The data is actually taken from the daily mean SYN product. This has been corrected in the text
* * *
*Fig. 5b and c:: It is commendable that the authors have partly explained why all the in situ cirrus categories have more SW CRE than LW CRE (due to low clouds). These cirrus are typically having lower optical depth and thus lower SW and LW CRE (Fig. 5c), with TOA LW CRE > SW CRE (e.g. Fu, 2008, Fig. 4; Hong and Liu, 2015, J. Climate). But after removing the low clouds in Fig. 5c, in situ cirrus still have a net CRE < zero, whereas other studies infer positive values. For example, for cirrus OD < 3.6 and cloud top pressure <440 mb, the net CRE reported by Chen et al. (2000, J. Clim.) was positive, as was also true for Hartman et al. (1992, J. Clim.) for cirrus OD <9.4. Based on the ECHAM6 GCM, the global average net CRE of cirrus clouds is +5.7W/m2 (Gasparini and Lohmann, 2016, JGR). The proposed technical note ap-*

[Figure]

pears to be at variance with the literature in regards to the overall sign of the net cirrus *forcing, and this discrepancy should be addressed. Note that the calculations in Fu are for the equator during an equinox when the sun is highest in the sky, which maximizes the SW CRE.*

**Reply**: Thank you for drawing our attention to these studies. We have gone back to look at the CRE data again. The average net synoptic CRE is slightly positive, at around 3Wm$^{-2}$ (similar to previous work), when removing any liquid clouds, which is more consistent with methods in previous work. The overall cirrus net CRE is still negative, but this is mostly due to the strong negative effect on the frontal and convective regimes. We suspect that the difference to previous work most probably comes come the different pressure levels used to separate out low cloud. Hartmann et al, 1992 used 440 hPa, whereas we have used 550 hPa. This results in a slightly less positive LW CRE, contributing to the overall smaller CRE. A clause has now been added noting that the net synoptic CRE is now positive (P17L25).

*Page 14, line 16:: "As seen in previous studies, the net cloud radiative forcing (CRE) is negative". Yes, but this paper is about cirrus clouds, and their net CRE is positive. Please cite these "previous studies" that pertain to cirrus clouds. One study by Chen et al. (2000, J. Clim.) was cited in the Introduction and could be cited again here. As noted above, Chen et al. (2000) and Hartman et al. (1992) found that for cirrus OD <3.6 or 9.4, respectively, their net CRE is positive.*

**Reply**: This sentence was intended to point out that the total CRE is negative and has now been amended.

Minor Comments

*Page 2, line 2:: Remove "and" from this sentence.*
**Reply**: Done

***Page 2, line 12::*** *Comma not needed*
**Reply**: Amended
* * *
***Page 2, line 18::*** *"Existing Classifications" should be given a sub-header value of 3.1.*
**Reply**: Amended
* * *
***Page 3, line 27::*** *determines => determining?*
**Reply**: Amended
* * *
***Figure 4::*** *Please label all the panels as a, b and c. Also, what do the 3 horizontal lines indicate in the middle-regions of Fig. 4b and 4c?*
**Reply**: Amended. The caption now states that the grey lines are gridlines.
* * *
***Page 12, line 15::*** *Fig. 5a does not show RFO; please clarify. Also, "frontal convective regimes" => "frontal and convective regimes"? Based on Fig. 5b, frontal and convective regimes appear to account for slightly > 12%.*
**Reply**: Corrected to refer to Fig. 2, percentages and missing "and" amended.
* * *
***Page 13, line 8::*** *"once" => "until"?*
**Reply**: This sentence has been re-written for clarity.

**References**

Hartmann, D. L., Ockert-Bell, M. E., and Michelsen, M. L.: The Effect of Cloud Type on Earth's Energy Balance: Global Analysis, J. Climate, 5, 1281–1304, doi:10.1175/1520-0442(1992)005<1281:TEOCTO>2.0.CO;2, 1992.

Heymsfield, A. J., Krämer, M., Luebke, A., Brown, P., Cziczo, D. J., Franklin, C., Lawson, P., Lohmann, U., McFarquhar, G., Ulanowski, Z., and Van Tricht, K.: Cirrus Clouds, Meteorological Monographs, 58, 2.1–2.26, doi:10.1175/AMSMONOGRAPHS-D-16-0010.1, 2017.

Medeiros, B. and Stevens, B.: Revealing differences in GCM representations of low clouds, Climate Dyn., 36, 385–399, doi:10.1007/s00382-009-0694-5, 2011.

Nam, C. C. W. and Quaas, J.: Geographical versus dynamically defined boundary layer cloud regimes and their use to evaluate general circulation model cloud parameterizations, Geophys. Res. Lett., doi:10.1002/grl.50945, 2013.

Zhang, S., Wang, M., Ghan, S. J., Ding, A., Wang, H., Zhang, K., Neubauer, D., Lohmann, U., Ferrachat, S., Takeamura, T., Gettelman, A., Morrison, H., Lee, Y., Shindell, D. T., Partridge, D., Stier, P., Kipling, Z., and Fu, C.: On the characteristics of aerosol indirect effect based on dynamic regimes in global climate models, Atmos. Chem. Phys., 16, 2765–2783, doi:10.5194/acp-16-2765-2016, 2016.

———————————————————————